# Wandering albatrosses exert high take-off effort only when both wind and waves are gentle

Leo Uesaka[1,2]*, Yusuke Goto[1,3,4], Masaru Naruoka[5], Henri Weimerskirch[4], Katsufumi Sato[1], Kentaro Q Sakamoto[1]

[1]Atmosphere and Ocean Research Institute, The University of Tokyo, Kashiwa, Japan; [2]Information and Technology Center, The University of Tokyo, Kashiwa, Japan; [3]Graduate School of Environmental Studies, Nagoya University, Furo, Japan; [4]Centre d'Etudes Biologiques de Chize (CEBC), UMR 7372 CNRS, Université de La Rochelle, Villiers-en-Bois, France; [5]Aeronautical Technology Directorate, Japan Aerospace Exploration Agency (JAXA), Chofu, Japan

*For correspondence:
leo-u@g.ecc.u-tokyo.ac.jp

Competing interest: The authors declare that no competing interests exist.

**Abstract** The relationship between the environment and marine animal small-scale behavior is not fully understood. This is largely due to the difficulty in obtaining environmental datasets with a high spatiotemporal precision. The problem is particularly pertinent in assessing the influence of environmental factors in rapid, high energy-consuming behavior such as seabird take-off. To fill the gaps in the existing environmental datasets, we employed novel techniques using animal-borne sensors with motion records to estimate wind and ocean wave parameters and evaluated their influence on wandering albatross take-off patterns. Measurements revealed that wind speed and wave heights experienced by wandering albatrosses during take-off ranged from 0.7 to 15.4 m/s and 1.6 to 6.4 m, respectively. The four indices measured (flapping number, frequency, sea surface running speed, and duration) also varied with the environmental conditions (e.g., flapping number varied from 0 to over 20). Importantly, take-off was easier under higher wave conditions than under lower wave conditions at a constant wind speed, and take-off effort increased only when both wind and waves were gentle. Our data suggest that both ocean waves and winds play important roles for albatross take-off and advances our current understanding of albatross flight mechanisms.

## eLife assessment

This **fundamental** study advances our understanding of seabird responses to environmental conditions, with implications for movement ecology, flight biomechanics, animal foraging, and bioenergetics. Animal-borne data-loggers are used to generate a **compelling** high quality dataset on animal movement and environmental conditions. The study will interest ornithologists, comparative bio-mechanists, ocean ecologists and those interested in technological advances in animal sensors.

## Introduction

Various oceanic environmental factors affect the ecology of marine animals. Predicted climate changes suggest increases in extreme climatic events (such as cyclones). Thus, evaluating individual relationships between each environmental factor and marine animal behaviors is urgent for marine ecological conservation, especially for top predators that significantly impact the entire ecosystem. However, there are potential limitations: direct measures of marine animals empirical environmental data are nearly impossible due to the spatiotemporal gaps in the observation network of the open

**eLife digest** Wandering albatrosses are large seabirds with one of the most impressive wingspans found in the animal kingdom. While they spend most of their time efficiently gliding above the waves, they do have to regularly land on sea to snatch their prey. To resume flight, the birds turn into the wind and flap their wings as they run on the surface of the ocean; this causes their heart to beat three to four times faster than normal. In contrast, flying barely leads to a change in pulse rate compared to rest. As for many other marine birds, sea take-offs therefore represent one of the major energy costs that albatrosses face when out foraging.

Scientists have long assumed that the amount of effort required for this manoeuvre depends on factors such as wind speed and, potentially, the height of the waves. However, this is difficult to establish for sure because direct information about the environment that a bird faces as it takes off is rarely available. Often, the best that researchers can do is to reconstruct this data based on global weather patterns, ocean climatic models or evidence collected from nearby locations.

To address this problem, Uesaka et al. devised innovative ways to use data from animal-borne sensors. They equipped 44 albatrosses with these instruments and recorded over 1,500 hours of foraging sea trips. Wind parameters such as speed and direction were estimated based on the animals' flying paths, and wave height calculated from their floating motion. Sensor data also gave an insight into the energy cost of each take-off, which was estimated based on four parameters (running duration, running speed, number of wing flaps, and flapping frequency).

The analyses confirmed that albatrosses take off into a headwind, with stronger winds reducing the amount of effort required. However, wave height also had a profound impact, suggesting that this parameter should be included in future studies. Overall, the birds flapped their wings less and ran on the surface of the water for shorter amounts of time when the wind was strong, or the waves were high. Even with weak winds, take offs were easier when waves were taller, and they were most costly when both the sea and wind were calm.

The work by Uesaka et al. helps to capture how environmental factors influence the energy balance of albatrosses and other marine birds. As ocean weather patterns become more volatile and extreme climate events more frequent, such knowledge is acutely needed to understand how these creatures may respond to their changing world.

ocean (*Ardhuin et al., 2019*; *Villas Bôas et al., 2019*). Various environmental parameters (such as ocean wind, waves, and sea surface temperature) are assumed to be important factors affecting the movement and foraging of flying seabirds (*Dunn, 1973*; *Haney et al., 1992*; *Adams and Navarro, 2005*; *Nevitt et al., 2008*; *Suryan et al., 2008*). Previous research has revealed that many interesting seabird behaviors correlate with the ocean environment. However, the environmental data largely rely on ocean climatic models (*Padget et al., 2019*; *Weimerskirch and Prudor, 2019*; *Clay et al., 2020*; *Clairbaux et al., 2021*), as in situ observation data are limited and often collected a long distance from the bird. For example, records are collected at the colony island or using the nearest government observation point (*Kogure et al., 2016*; *Yamamoto et al., 2017*). Therefore, interpreting the data is difficult when doubts exist on whether birds actually experienced the same environmental conditions, making any conclusions conservative estimates only (*Clay et al., 2020*; *Clairbaux et al., 2021*). For instance, although winter cyclones in the North Atlantic can induce mass seabird mortality, revealing the small-scale behavioral responses which lead to mortality is almost impossible with the spatiotemporal limits of thermodynamic modeling data (*Clairbaux et al., 2021*).

Seabird take-off may be affected by the surrounding environment (*Clay et al., 2020*) but has never been effectively investigated. Notably, behaviors with short timeframes (such as take-offs) require localized environmental data on spatiotemporally small scales, which is difficult to obtain, even using mathematical weather models. Many procellariiformes have special flight techniques that use vertical wind shear, called dynamic soaring (*Rayleigh, 1883*; *Richardson, 2011*), while take-off requires a large amount of energy (*Sakamoto et al., 2013*) owing to vigorous flapping (*Sato et al., 2009*; *Sakamoto et al., 2013*) and sea surface running to reach the velocity to initiate take-off (*Sato et al., 2009*). Previous research revealed the heart rate of the largest seabird, wandering albatross (*Diomedea exulans*), drastically increases at the moment of take-off reaching three to four times the basal heart

rate (*Weimerskirch et al., 2000*). After take-off, the tachycardia progressively decreases during flight (*Weimerskirch et al., 2000*), the flying heart rate is close to the basal rate of a resting bird on the nest. Therefore, the high energy expenditure associated with take-off strongly influences the total energy expenditure of wandering albatross during the foraging trip, unlike flight duration or distance (*Shaffer et al., 2001a*). Thus, take-off is one of the most important behaviors in the daily energy budget of flying seabirds in the open ocean. Understanding the relationship between take-off and the ocean environment is critical for estimating the future climate change effects on the life history of seabirds (*Weimerskirch et al., 2012*).

Previous research has partially identified the role of wind conditions on take-off when investigating general flight tactics of seabirds (*Kogure et al., 2016*; *Clay et al., 2020*). For example, the flapping effort of the European shag (*Gulosus aristotelis*) at take-off decreases as wind speed increases (*Kogure et al., 2016*). However, a comprehensive understanding of take-off has not been achieved as other environmental parameters, such as waves, have not been investigated. Ocean waves potentially affect take-off efforts because seabirds usually run on the ocean surface as they take-off (*Norberg and Norberg, 1971*; *Sato et al., 2009*). Additionally, the ocean surface slope is a key factor in creating complicated wind patterns immediately above the surface, which may affect the flight tactics of procellariiformes (*Bousquet et al., 2017*).

In this study, we devised a new approach to estimate the empirical local environmental conditions using seabird dynamic motion records without the aid of either mathematical weather models or observational data. The recent development of animal-borne recorders has been remarkable (*Wilmers et al., 2015*). It is now possible to deploy various sensors on animal-borne recorders, to generate a new field of oceanography: ocean observations using animal-borne sensors (*Harcourt et al., 2019*; *McMahon et al., 2021*). Many studies have reported that marine environmental data can be collected using highly mobile marine animals such as pinnipeds, sea turtles, and seabirds (*Charrassin et al., 2002*; *Charrassin et al., 2008*; *Roquet et al., 2014*; *Doi et al., 2019*). Furthermore, unlike direct measurement by deploying sensors (e.g., thermometers), indirect techniques to observe the physical environment via the dynamic animal motion records (*Yoda et al., 2014*; *Yonehara et al., 2016*; *Goto et al., 2017*; *Sánchez-Román et al., 2019*; *Uesaka et al., 2022*) generated using the Global Navigation Satellite System (GNSS) have been developed recently. GNSS is now able to record animal

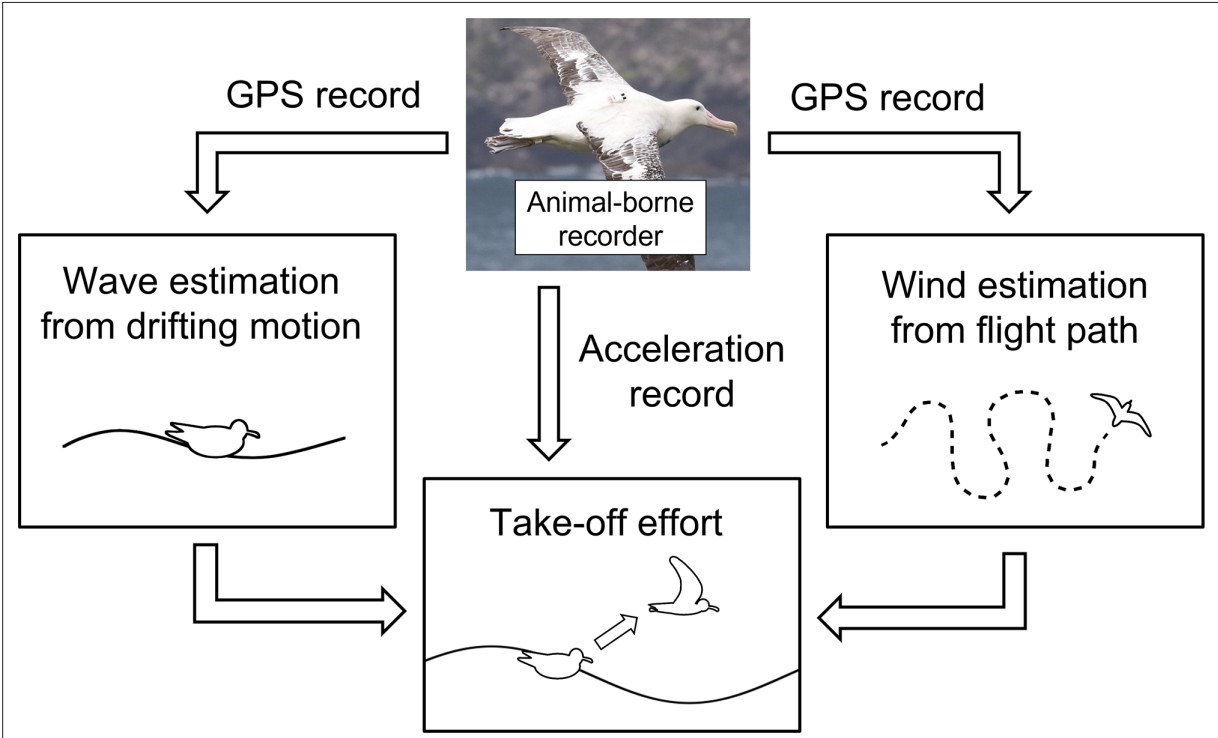

**Figure 1.** Conceptual framework of the study estimating environmental conditions experienced by studied individual.

position and movement every second (or even in sub-second scales) using very small animal-borne recorders. These newly developed techniques using animal-borne recorders should compensate for previous observational gaps in oceanic data, especially in the open ocean and polar regions where our access and deployment of observation equipment is complicated. The environmental variables obtained directly from free ranging animals provide the localized environmental conditions they experience. Seabirds are one of the most enthusiastically studied oceanic species because of their high mobility and adaptability to both air and water. Methods involving wind and wave observation, using GNSS data regarding the flight paths and floating motions of seabirds on the sea surface, are well developed and have potential applicability in future studies (*Yonehara et al., 2016*; *Goto et al., 2017*; *Uesaka et al., 2022*).

Wandering albatrosses were investigated because their habitat includes the Subantarctic (30°S–60°S), where the ocean is annually rough (*Suryan et al., 2008*) causing their flight behaviors to be largely influenced by the ocean conditions (*Richardson, 2011*; *Weimerskirch et al., 2012*). Furthermore, previous studies have revealed the foraging area has shifted southward annually with the polar shift of the westerly wind pattern (*Weimerskirch et al., 2012*). Considering the enormous cost of take-off (*Weimerskirch et al., 2000*; *Shaffer et al., 2001a*), studying their response to various environmental conditions is essential for us to estimate the impacts of climate change on the life history of seabirds. We aim to estimate the physical environmental conditions (ocean winds and wave heights) experienced by wandering albatrosses as they take-off by utilizing the dynamic motion records to evaluate the effects of wind and wave conditions on take-off dynamics (*Figure 1*). Procellariiformes, like many seabird species, require extensive limb motion for take-off which is not limited to flapping behavior. Therefore, the evaluation of take-off effort involves both surface running and flapping behaviors. The wandering albatross individuals were tagged using recorders that include both global navigation satellite system (GNSS), specifically the global positioning system (GPS), and acceleration sensors with high time resolutions.

## Results

### Trip data

We obtained 1477 hr from 44 wandering albatrosses in 2019 (*N* = 21, 623 hr) and 2020 (*N* = 23, 854 hr). Two types of recorders with different battery sizes were used. The mean recording time of the trip data and the standard deviation (SD) was 9.5 ± 1.3 hr for the small battery recorders and 59.7 ± 9.6 hr for the large battery recorders. The albatross sex ratio was balanced between years and recorder type (*Supplementary file 1*).

The absolute value of the GPS horizontal velocity revealed 703 take-offs from 1477 hr of trips. A total of 453 out of 703 take-offs were followed by more than 5 min of flight. For each flight, the wind speed and direction were estimated using the flight path (*Yonehara et al., 2016*). A total of 299 take-offs occurred after more than 15 min of floating time. Wave heights were estimated for each of the 299 take-offs using the floating motions (*Uesaka et al., 2022*). For 185 take-offs, we estimated the wind and wave conditions in combination.

### Environmental conditions at the take-off moment

Of the 453 estimated wind parameters, 26 were unreliable based on the Akaike information criterion (AIC) comparison and were not included in the analysis. The remaining 427 results revealed wind speeds of 6–8 m/s were most frequently experienced by taking-off wandering albatrosses (*Figure 2A*). Mean ± SD of the estimated wind speed was 6.5 ± 2.7 m/s, and the maximum and minimum wind speeds were 15.4 and 0.7 m/s, respectively. Winds blowing from west to east were frequently observed (*Figure 2B*). This result is consistent with the prevalence of westerlies around the wandering albatross breeding colony (*Nicholls et al., 1997*; *Weimerskirch et al., 2015*).

Ocean waves were estimated using all 299 take-offs after more than 15 min of floating time to calculate the significant wave height. The most frequently experienced wave heights ranged from 2.5 to 3.0 m at the take-off moment (*Figure 2C*) and the mean ± SD was 3.0 ± 0.8 m. The minimum and maximum wave heights were 1.6 and 6.4 m, respectively. Like wind direction, the wave direction (coming from) had a west bias due to the westerlies (*Figure 2D*).

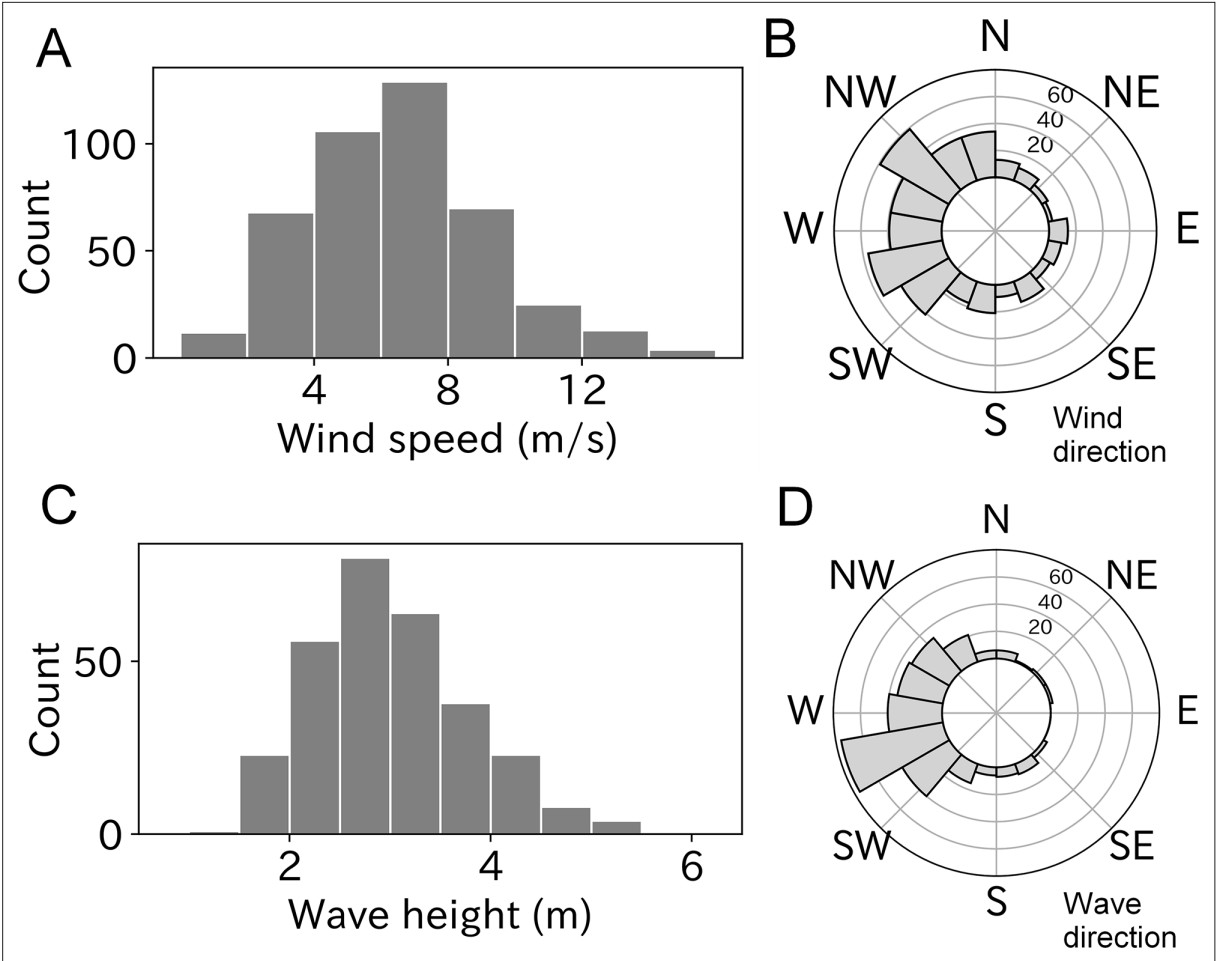

**Figure 2.** Wind and wave conditions experienced by taking-off wandering albatrosses. (**A, B**) Histogram of windspeed and wind direction ($n = 427$). (**C, D**) Histogram of wave height and wave direction ($n = 299$).

## Take-off properties

To quantify the take-off effort, we calculated four parameters: running duration, running speed, flapping number, and flapping frequency from the acceleration records obtained at the moment of take-off. Mean ± SD running duration of wandering albatross was 5.1 ± 1.5 s with a range from 1.1 to 11.7 s (*Figure 3A*). The mean value for males was slightly lower than that for females (*Figure 3—figure supplement 1A*), however, the difference was not significant (M: 5.0 ± 1.5 s, F: 5.2 ± 1.5 s, $p = 0.10$, Mann–Whitney *U*-test). The albatross running speed mean value ± SD was 6.5 ± 1.6 m/s (*Figure 3B*). Male birds had slightly higher speeds than females (M: 6.7 ± 1.5 m/s, F: 6.3 ± 1.6, $p < 0.01$, *Figure 3—figure supplement 1B*). Running duration and speed significantly correlated (Pearson's $r = 0.57$, $p < 0.01$, *Figure 3—figure supplement 2*). The linear regression slope (with a fixed intercept of zero) was 1.23 m/s². The slope can be interpreted as the running wandering albatross acceleration.

The flapping number, that is, the number of wing flaps after the running phase, was estimated using the dorsoventral acceleration. The mean flapping number was 4.3 times with a range from zero to over 20 times (*Figure 3C*). Take-offs without flapping after the running phase were frequently observed (33.3%). Conversely, continuous flapping above 20 times were also occasionally observed, which corresponds to a lengthy flapping duration (8 s <) after take-off, considering the flapping frequency of wandering albatross (2.5–3.0 Hz). There was no significant difference in flapping number between the sexes ($p = 0.22$, Mann–Whitney *U*-test, *Figure 3—figure supplement 1C*). The mean ± SD flapping frequency was 2.55 ± 0.29 Hz, and most ranged from 2 to 3 Hz (*Figure 3D*). However, some flapping frequency results were outside the detection range (1.8–4 Hz) and not included in our analysis. Therefore, the sample size of the flapping frequency used in our analysis was 669. There was

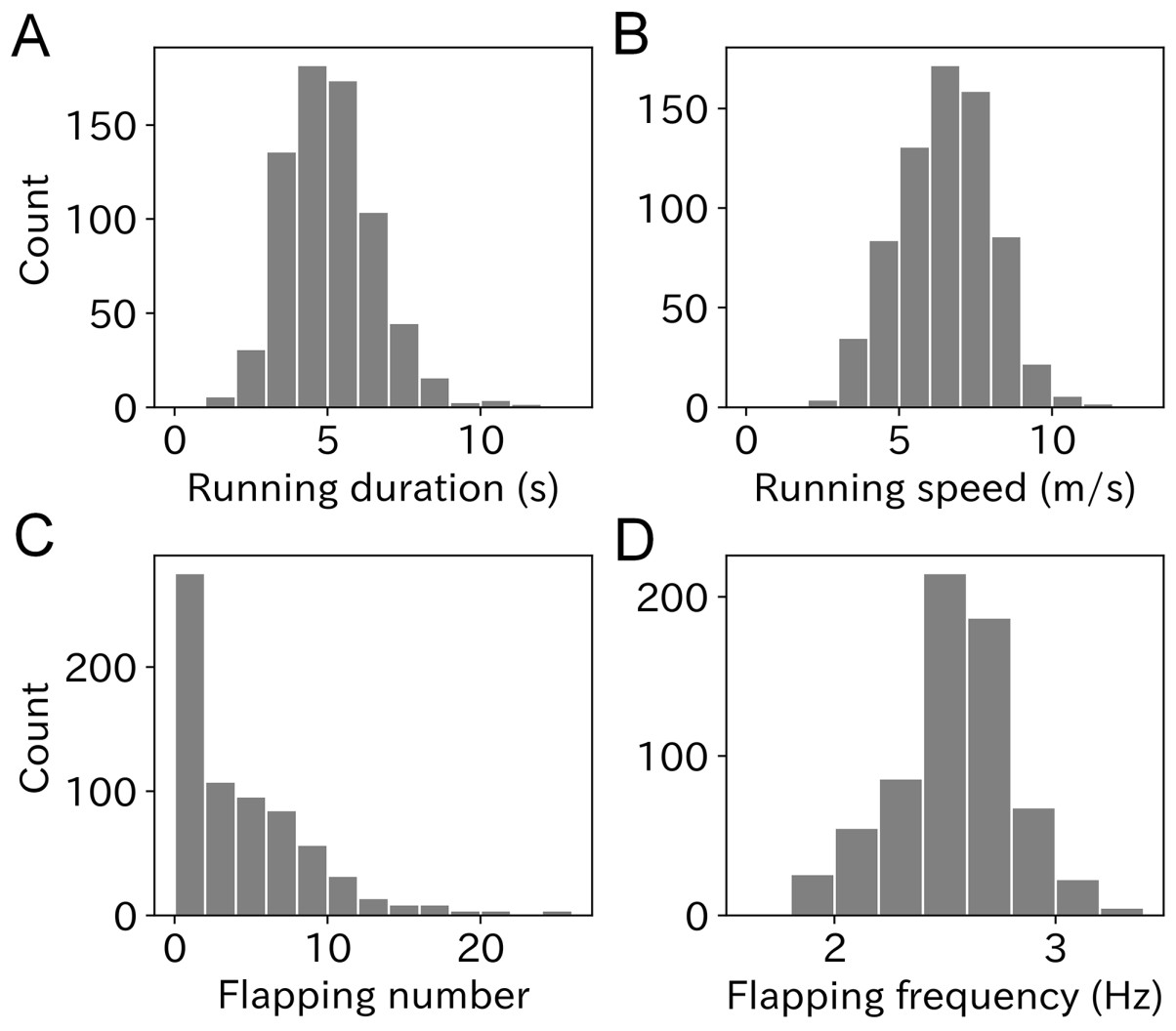

**Figure 3.** Histogram of (**A**) running duration, (**B**) running speed, (**C**) flapping number, and (**D**) flapping frequency. Graphs A, B, and C are composed of 703 samples and graph D is composed of 669 samples.

The online version of this article includes the following figure supplement(s) for figure 3:

**Figure supplement 1.** Sexual difference of (**A**) running duration, (**B**) running speed, (**C**) flapping number, and (**D**) flapping frequency.

**Figure supplement 2.** Running duration and running speed had significant correlation ($r = 0.57$, $p < 0.01$, $n = 703$).

no significant difference in flapping frequency between the sexes ($p = 0.18$, Mann–Whitney $U$-test, *Figure 3—figure supplement 1D*).

## Environmental effects on take-off parameters

The take-off directions were compared with the wind direction estimated from the flight path after take-off. Wandering albatrosses tended to take-off with headwinds ($p < 0.01$, $v$-test) (*Figure 4*). However, the cruising direction (moving direction from the take-off point to the bird location after 5 min) did not correlate with headwind direction. The mean ± SD air speed of wandering albatrosses at the end of the running phase (lift-off moment from the sea surface) calculated using the running speed, wind speed, and relative take-off direction was 12.2 ± 3.1 m/s.

The relationships between each take-off parameter (running duration, running speed, flapping number, and flapping frequency) with environmental conditions (wind speed and wave height) were tested using linear mixed models (LMM). The running duration required for wandering albatross take-off significantly decreased as wind speed and wave height increased (*Figure 5*). Similarly, the running speed was significantly lower under stronger wind and higher wave conditions. Wandering

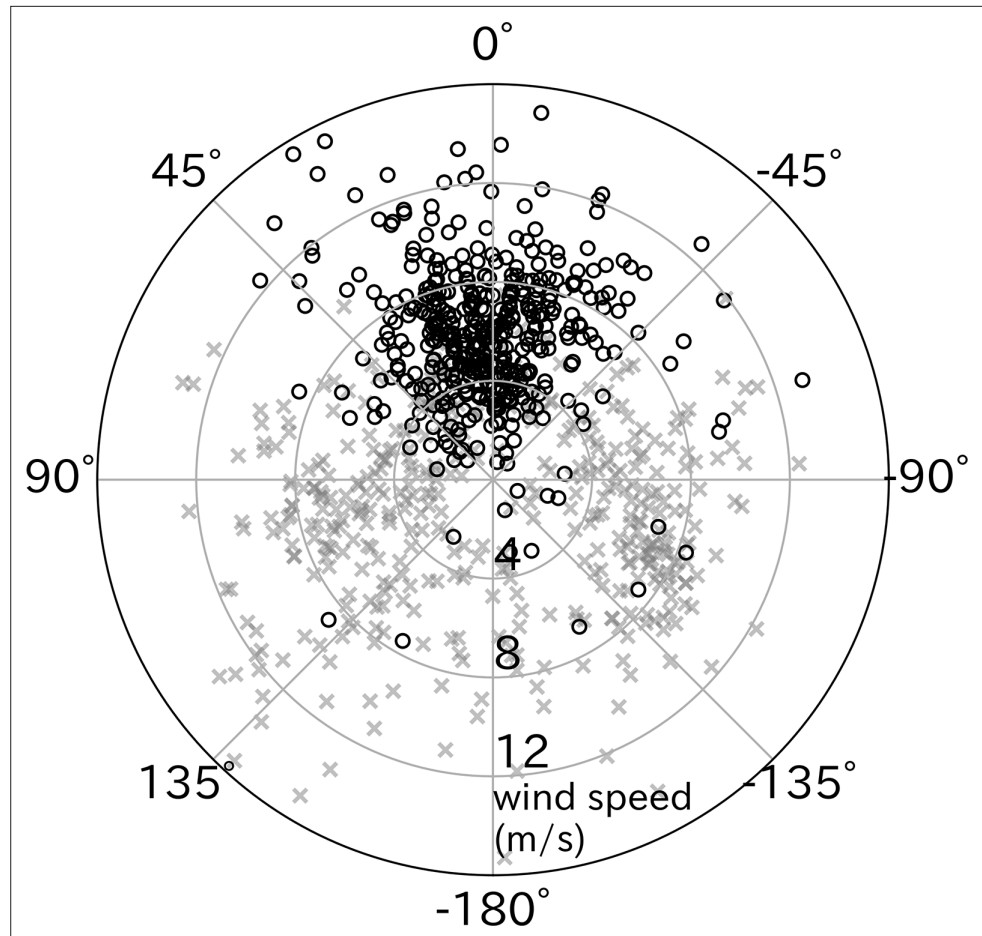

**Figure 4.** Effect of wind direction on wandering albatross take-off. Relative take-off direction to wind direction (black circles, *n* = 427) significantly distributed around 0° (headwind), in contrast to cruising direction relative to the wind (gray x-mark, *n* = 427). The radial axis represents the wind speed.

albatrosses tend to flap fewer times under stronger wind conditions. Conversely, wandering albatrosses can flap over 20 times in weak wind conditions, although the flapping number in weak wind conditions varies greatly. There is also a declining trend in the flapping number with wave height. Albatross take-offs in wave heights below 2 m always require flapping. The flapping frequency was lower as the wind speed and wave height increased, however, the trend with higher wave heights remains unclear. The LMM results are provided in *Table 1*.

### Independent effect of wind and waves on take-off

Although some ocean wave components are generated by ocean winds, the correlation between the wind speed and wave height is not consistent. Some of the albatross take-offs involved information on both wind speed and wave height. Therefore, we evaluated the respective effects of wind and waves on wandering albatross take-offs. The correlation between wind speed and wave height was not strong (*r* = 0.27, p < 0.01). Some take-offs were performed in weak winds but high wave conditions or the opposite conditions (*Figure 6A*). Take-off conditions were divided into four environmental categories using the peak value, which were 6.0 m/s (wind speed) and 2.8 m (wave height). The categories comprised: 48 samples (weak wind low wave: WL), 33 samples (weak wind high wave: WH), 27 samples (strong wind low wave: SL), and 77 samples (strong wind high wave: SH). The running duration varied significantly between the four categories (p < 0.01, Kruskal–Wallis test). The mean running duration in the WL conditions was 6.0, which was the longest of the four categories (*Figure 6B*). Relatively long running (of over 6 s) mainly occurred in WL conditions, and the running duration decreased with the wind speed or wave height (*Figure 6—figure supplement 1*). Similar results were obtained

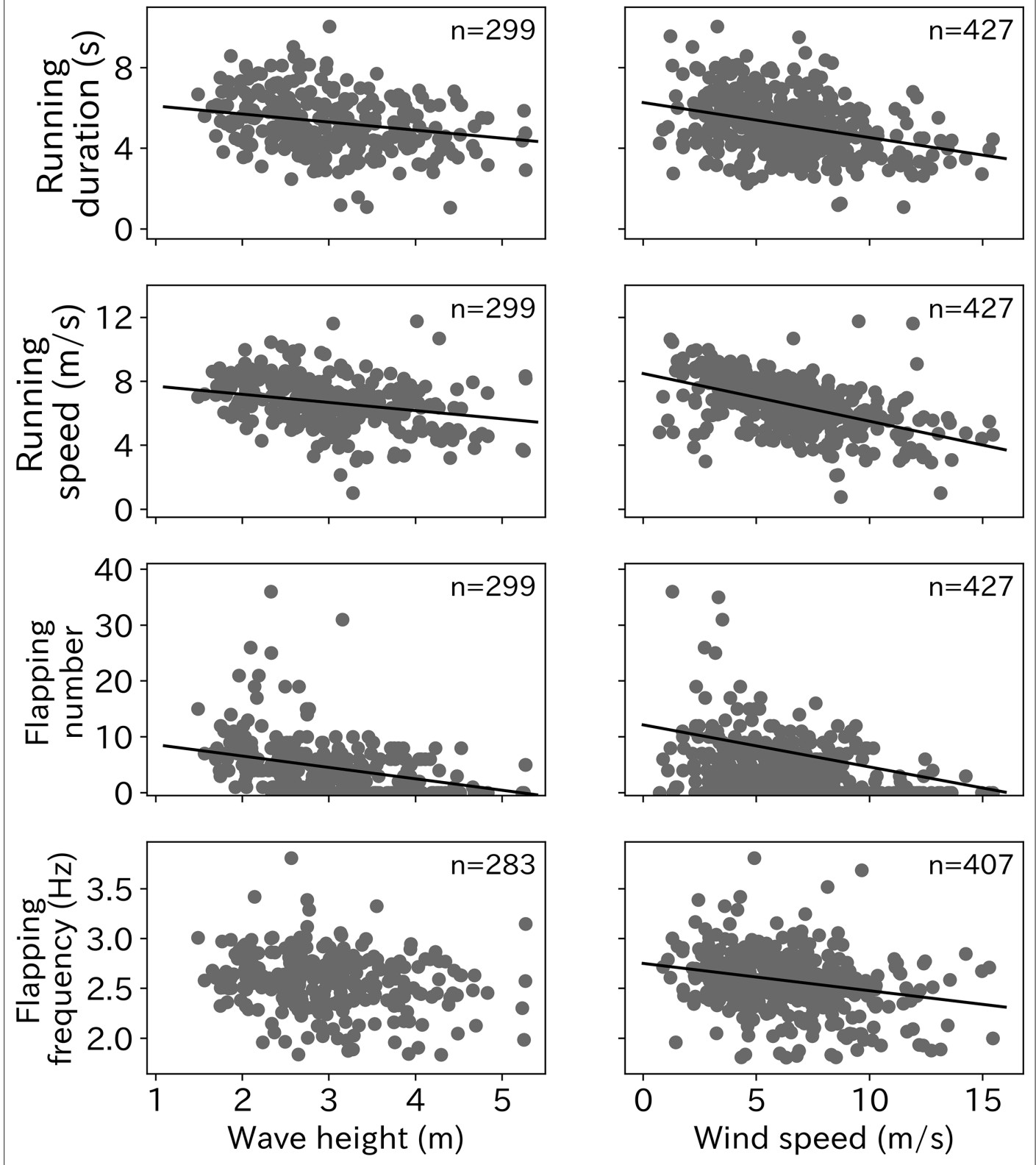

**Figure 5.** Environmental effect on take-off. Effort for the take-off (running duration, running speed, flapping number, and flapping frequency) significantly decreased as wave height and wind speed increased (p < 0.01) except the relationship between flapping frequency and wave height (p = 0.026). Solid line shows the linear regression line determined from the LMM and the number at the right top corner on each graph shows the sample sizes.

**Table 1.** Result of Akaike information criterion (AIC) and p values from LMM estimating the environmental effect on take-off behaviors.

The best models are shown in bold.

| Response variables | N | Explanatory variables | AIC | p value (Chi square) |
|---|---|---|---|---|
| Running duration | 427 | Null | 1505.2 | |
| | | Wind speed | 1464.4 | $6.1 \times 10^{-11}$ |
| Running duration | 299 | Null | 1052.6 | |
| | | Wave height | 1042.0 | $4.0 \times 10^{-4}$ |
| Running speed | 427 | Null | 1584.5 | |
| | | Wind speed | 1480.0 | $2.2 \times 10^{-16}$ |
| Running speed | 299 | Null | 1092.9 | |
| | | Wave height | 1077.8 | $3.4 \times 10^{-5}$ |
| Flapping number | 427 | Null | 2867.0 | |
| | | Wind speed | 2834.4 | $4.0 \times 10^{-9}$ |
| Flapping number | 299 | Null | 1798.6 | |
| | | Wave height | 1773.5 | $2.0 \times 10^{-7}$ |
| Flapping frequency | 407 | Null | 138.4 | |
| | | Wind speed | 115.6 | $6.4 \times 10^{-7}$ |
| Flapping frequency | 283 | Null | 70.1 | |
| | | Wave height | 67.2 | 0.026 |

for both running speed and flapping number. Take-offs involving over 30 flaps mainly occurred in WL conditions. Flapping frequency did not significantly vary between the four categories (p = 0.06).

The variance inflation factor (VIF) of wind speed and wave height was 6.86, which did not exceed the general threshold of 10 (*Dormann et al., 2013*). Among the LMM results, models including wind speed, wave height, and the interaction used the smallest AIC for all take-off parameters (*Table 2*). However, the difference between the lowest and the second lowest AIC was below two for running speed, flapping number, and flapping frequency. The running duration simulation using the estimated coefficient shows that even under weak wind conditions (2 m/s), running duration decreases from 8 to 4 s as the wave height increases. Conversely, low values were maintained under strong wind conditions (8 m/s) regardless of the wave height (*Figure 7*). Similarly, the running speed decreased from 9 to 6 m/s as the wave height increased, regardless of the wind strength. The flapping number followed the same trend. Conversely, the flapping frequency did not decrease as the wave height increased.

## Discussion

Although observational networks in the ocean are under development and mathematical weather modeling accuracy is increasing, they remain unable to accurately estimate the surrounding

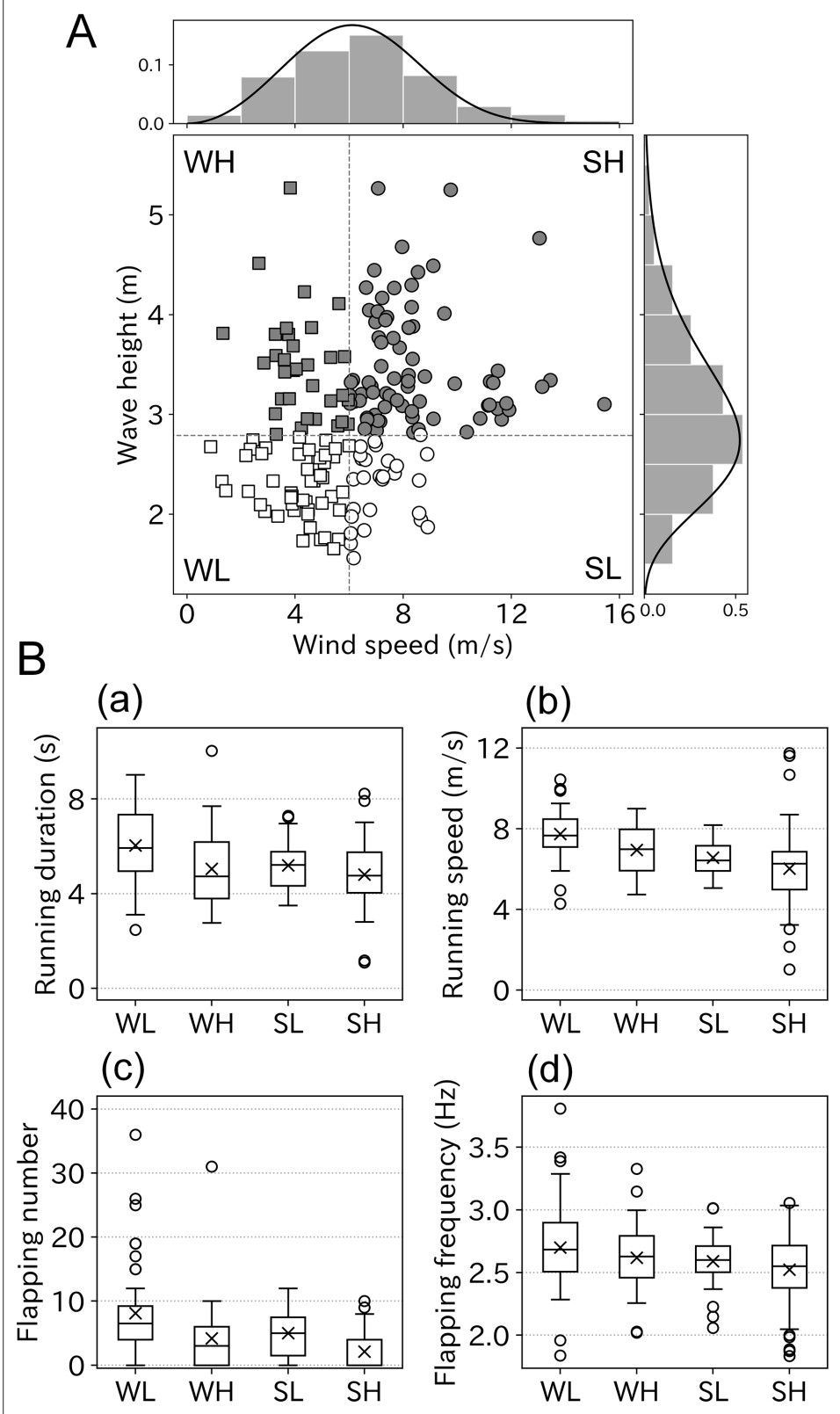

**Figure 6.** Take-off conditions divided into four environmental categories. (**A**) Correlation between wind speed and wave height was weak (*r* = 0.27, *n* = 185). Bar charts and solid lines written above and right of the scatter plot are normed histograms of wind speed, wave height, and curve fitted lines. Based on the peak value of fitted lines scatter plots were divided into four categories, WL: weak wind low wave (open square, n = 48), WH: weak wind

*Figure 6 continued on next page*

*Figure 6 continued*

high wave (filled square, n = 33), SL: strong wind low wave (open circle, n = 27), and SH: strong wind high wave (filled circle, n = 77). (**B**) Take-off effort comparison among four categories (a: running duration, b: running speed, c: flapping number, and d: flapping frequency). Cross mark indicates the mean value.

The online version of this article includes the following figure supplement(s) for figure 6:

**Figure supplement 1.** Take-off effort (gray scale) in relation to wind speed and wave height.

environment of marine animals at small scales. Here, we demonstrated that environmental variables estimated using individual animal recorders provide valuable new insight into locomotor behavior when spatiotemporal scale and accuracy of mathematical weather models and observational networks are too broad for the research. In this study, we provided details on how seabird take-offs are affected by wind and waves.

## Seabird take-offs using accelerometers

We quantified the running behavior of seabirds at the moment of take-off, which is the most energy-consuming behavior for soaring seabirds (*Weimerskirch et al., 2000*; *Shaffer et al., 2001a*; *Sakamoto et al., 2013*). Previous studies have ascribed this large energy expenditure to the vigorous

**Table 2.** Results of Akaike information criterion (AIC) from LMM considering both wind speed and wave height as candidates affecting take-off behaviors.

The best models are shown in bold.

| Response variables | N | Explanatory variables | AIC |
|---|---|---|---|
| Running duration | 185 | Null | 655.81 |
| | | Wave height | 650.06 |
| | | Wind speed | 639.45 |
| | | Wind speed + Wave height | 637.81 |
| | | Wind speed + Wave height + Interaction | 631.02 |
| Running speed | 185 | Null | 699.62 |
| | | Wave height | 690.57 |
| | | Wind speed | 657.69 |
| | | Wind speed + Wave height | 655.76 |
| | | Wind speed + Wave height + Interaction | 655.49 |
| Flapping number | 185 | Null | 1129.4 |
| | | Wave height | 1117.7 |
| | | Wind speed | 1109.0 |
| | | Wind speed + Wave height | 1102.3 |
| | | Wind speed + Wave height + Interaction | 1101.7 |
| Flapping frequency | 175 | Null | 70.43 |
| | | Wave height | 70.96 |
| | | Wind speed | 52.25 |
| | | Wind speed + Wave height | 54.22 |
| | | Wind speed + Wave height + Interaction | 50.72 |

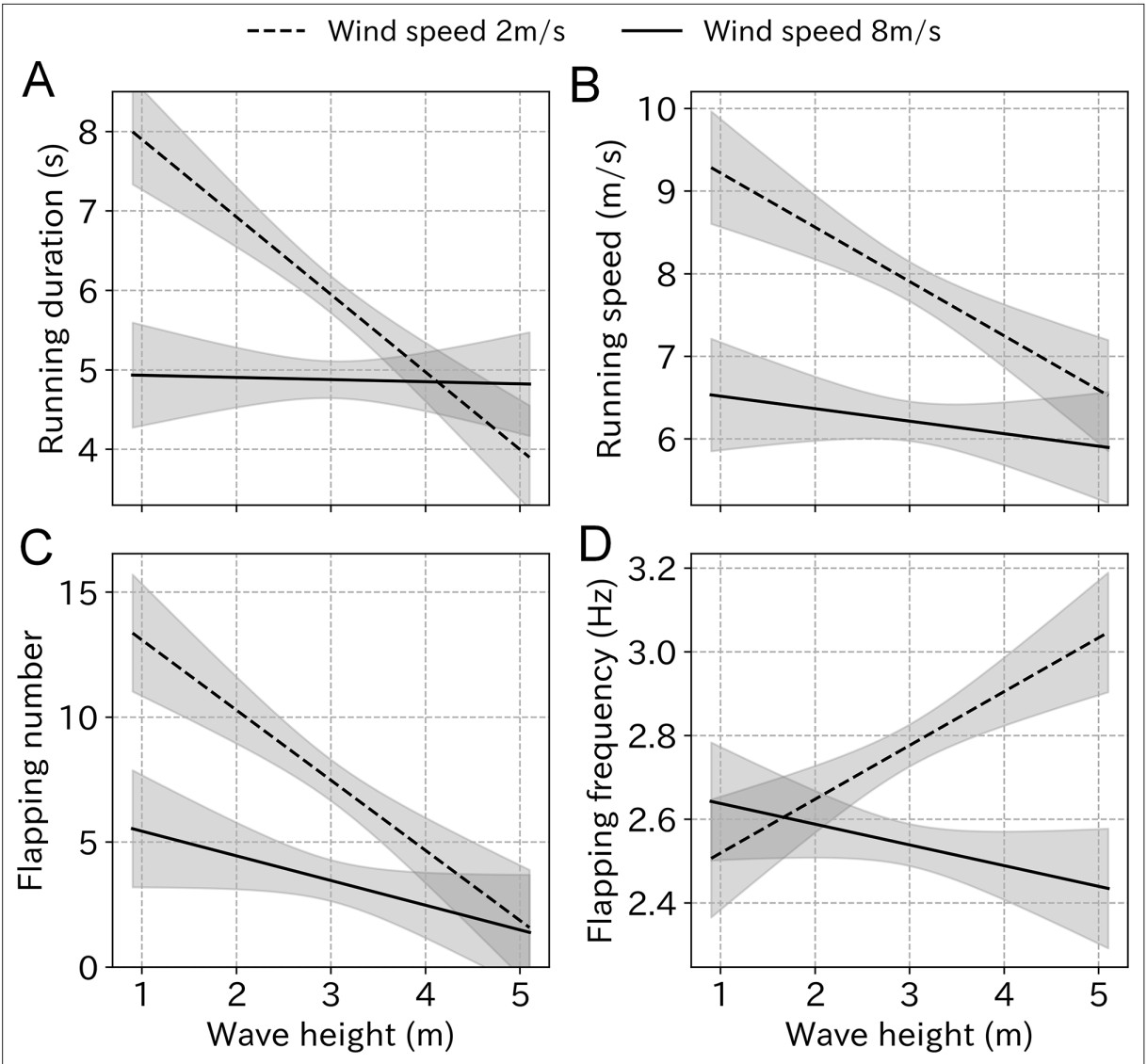

**Figure 7.** Take-off effort simulation using the estimated coefficient from LMM. (**A**) Running duration, (**B**) running speed, (**C**) flapping number, and (**D**) flapping frequency in response to the wave height change under weak wind (dashed line, 2 m/s) and strong wind (solid line, 8 m/s) conditions estimated from the LMM. Gray area represents 99% CI.

flapping required for take-off (*Shaffer et al., 2001a*; *Sato et al., 2009*; *Clay et al., 2020*). Indeed, the continuous flapping behavior, which is rare in cruising flight, was recorded even after the running phase of take-off in this study. However, we suggest that the running behavior should also entail a large cost in take-off because albatrosses have to reach a fast initial speed to lift off the sea surface by rapidly moving their hindlimbs for up to ~10 s in unfavorable conditions (as demonstrated in this study).

We provide a first attempt at detecting the running signal of seabird take-offs and construct a relatively simple algorithm (which can be easily applied to other species) using lateral acceleration. The running duration may increase or decrease by approximately 0.5 s depending on the algorithm configuration, such as smoothing parameters and threshold values. However, we focused on the relative changes in the running behavior in association with wind speed and wave height, and absolute value error is not a serious problem.

The flapping characteristics of wandering albatross during the running phase were also researched. However, the dorsoventral acceleration signal fluctuates during the running phase making it difficult to identify each flapping signal, even after applying the band-pass filter. Therefore, counting the

number of flaps immediately after the running phase was the only reliable parameter for evaluating flapping effort. *Sato et al., 2009* reported that the flapping frequency of wandering albatrosses at the moment of take-off is higher (2.9–3.4 Hz) than that of cruising flight (2.5–2.7 Hz). In our study, the flapping frequency after the running phase was not as high as *Sato et al., 2009* reported. Therefore, it is likely that wandering albatrosses undertake high-frequency flapping only during the running phase, as they lift off the sea surface. After lift-off (i.e., the running phase is completed), wandering albatrosses continue with a moderate flapping frequency until they reach a certain degree of flight stability. Simultaneous video records of flapping and running motions with acceleration records are required to separate the parameter estimates.

In-depth studies on seabird take-offs are just beginning with the aid of miniaturized animal-borne recorders with the main aim of understanding how seabirds flap their wings. However, in land birds (e.g., finches and doves) take-off requires a large contribution by hindlimbs (*Provini et al., 2012*) and the role of the hindlimb in take-off kinematics is as important as that of the wing (*Provini and Abourachid, 2018*). Therefore, it is highly likely that seabird take-offs also require a substantial contribution by the hindlimbs, and thus, further seabird hindlimb research is required. Our study provides the basic characteristics of wandering albatross running behavior, including running duration and speed.

## Take-off effort with environmental conditions

Our results demonstrate that wandering albatrosses can take-off in a variety of environmental conditions (wind speed: 0.7–15.4 m/s, wave height: 1.6–6.4 m). A previous study on wandering albatrosses identified the transition state from resting to flying tended to increase as the wind speed increased (*Clay et al., 2020*). Our results found some take-offs were performed under weak wind (2–4 m/s) conditions, suggesting wind speed is not the only parameter influencing flight decisions of wandering albatross, and that wave height should be included in future studies.

The results showed that the running and flapping behavior tended to decrease as the wind or wave conditions increased. Running duration decreased as either the wind speed or the wave height increased and peaked when both the wind and wave conditions were weak. The same trends existed in running speed and flapping number. Although optimum statistical models for each take-off parameter were determined using the AIC value, some models provided similar results to this model. For instance, the AIC difference in running speed between the best model and the second lowest AIC model was only 0.27. However, both models included wind speed and wave height as the explanatory variables, similar to the other take-off parameters, except flapping frequency. The purpose of constructing a linear model was to clarify whether the effects of wind and waves are independent. As long as both wind speed and wave height were included as explanatory variables in the model, they reduced the running and flapping behavior requirement. Therefore, we can conclude that both strong winds and high waves aid wandering albatross take-offs. The flapping frequency after the running phase was the only parameter that did not correlate with wave height (as identified using the LMM). However, we assume the flapping frequency during the running phase is more important. Future research needs to investigate the effects of wave height as wandering albatrosses need to climb up or run down the wave slope. Therefore, the flapping frequency during the running phase should be highly influenced by wave height.

## Contribution of strong wind and high waves to seabird take-off

The reduced running behavior and flapping times under strong wind conditions are simply described by the lift force mechanism which has been predicted by previous studies (*Kogure et al., 2016*; *Clay et al., 2020*). Seabirds need to gain lift force before take-off, and the magnitude of force is proportional to the square of the relative speed of the wings to the surrounding air (air speed) (*Vogel, 1983*). It has been anecdotally suggested that seabirds take-off into the wind (i.e., headwind), because stronger winds can produce a sufficiently large lift even before the ground speed of the seabird reaches the value required for flight. As a partial demonstration of this theory, a study on the European shag (*G. aristotelis*) *Kogure et al., 2016* found the take-off direction was significantly biased toward headwinds. Regarding soaring seabirds, only one study (*Clay et al., 2020*) on wandering albatrosses has confirmed a bias in take-off direction with wind direction. However, the authors acknowledge the limits in the mathematical weather model and GPS sampling resolutions and recognize the unreliability of small-scale responses to in situ variation in the atmosphere. Our study reveals wandering

albatrosses significantly tend to take-off into the wind, using robust fine scale data estimated from the flight records of wandering albatross. Moreover, there was no correlation between cruising and headwind direction, indicating that wandering albatrosses face the wind on take-off regardless of their destination. Our data are reliable as the empirical value provided is actually experienced by the albatross. Furthermore, by quantitatively evaluating the flapping and running effort, we demonstrate the theory of effortless take-offs by soaring seabirds in stronger wind conditions.

The mean air speed of wandering albatrosses at the end of the running phase was close to the average flight speed (approximately 15 m/s) (*Weimerskirch et al., 2002*), and similar to predicted best glide speeds (*Shaffer et al., 2001b*), indicating that wandering albatrosses gain sufficient lift at the end of the running phase and efficiently utilize ocean wind. Wind speed varies with altitude, therefore the wind blowing on the ocean surface must be smaller than the values estimated from the flight records of wandering albatross as they usually fly 3–12 m above the ocean surface (*Pennycuick, 1982*). Therefore, the calculated air speed is probably an overestimate when compared with the ocean surface. To compensate for the insufficient lift force gained during the running phase, wandering albatrosses flap their wings several times after the running phase. Therefore, the flapping number in weak wind conditions can exceed dozens before reaching stable flight.

The most important finding of our study is that the take-off effort estimated by the running behavior and number of flaps decreased not only with stronger winds but also with higher waves. While the role of ocean wind on flying seabirds has been well described (*Pennycuick, 2008*), how ocean waves influence the flight of seabirds remains largely unknown. However, many studies have reported the characteristic flight of soaring seabirds by tracking the ocean wave surface over long distances (*Pennycuick, 1982*; *Pennycuick, 2008*; *Richardson, 2011*; *Stokes and Lucas, 2021*), which even occurs in weak or no wind conditions (*Pennycuick, 1982*). Seabirds seem to be aided by atmospheric forces above the slope-like wave topography; the flight method using the shape of wave is called wave-slope soaring (*Richardson, 2011*). It is well recognized that air flows occur above ocean waves (*Buckley and Veron, 2016*; *Bousquet et al., 2017*). *Richardson, 2011* described the theoretical model of wave-slope soaring, where the flight mechanism of albatross is a combination of both dynamic soaring, which uses vertical wind shear above the ocean surface (~15 m), and wave-slope soaring, which uses the updraft caused by the wave topography. Thus, seabirds can continue to soar in weak wind conditions. Furthermore, mathematical analysis has revealed that the wave-induced updraft (even in windless conditions) can provide 60% of the transportation cost of a brown pelican (*Pelecanus occidentalis*, 2–3 kg), which is a wave-slope soarer (*Stokes and Lucas, 2021*). Thus, it is possible that the take-off effort by wandering albatross is also reduced by high waves. While qualitative field observations and mathematical demonstrations provide the only previous research on the role of waves on soaring seabirds, we experimentally demonstrated that ocean waves aid the most energy-consuming behavior, take-off. This finding helps future discussions on ocean topographical mechanisms affecting seabird flight.

The mechanism by which high waves aid wandering albatross take-off is not entirely clear. It is difficult to conclude a certain updraft is producing additional lift for wandering albatrosses, and it is also possible that there are other unresolved mechanisms. For example, a rough topographic surface can provide a favorable bump, like a slope or cliff to jump off into the air. Our results were restricted to wave height as the parameter of the ocean surface. Future research involving ocean surface steepness or wave frequency components will reveal the detailed mechanism of how waves facilitate seabird take-off behavior. In particular, ocean surface topography relies heavily on whether the dominant wave component is due to a swell (low-frequency waves propagated from a distance) or wind waves (high-frequency waves generated by local wind); moreover, this topography affects the wind pattern on the sea surface.

In conclusion, we revealed how the take-off effort of wandering albatross changes in various oceanic conditions. As take-off is one of the most energy-consuming behaviors that can dominate the total energy expenditure of a wandering albatross journey, these data will be of great value for considering how climate changes can alter the life of albatrosses. Future research, especially on albatrosses, should quantitatively evaluate the energy consumption of take-off with the wind and wave conditions. Currently, there is no major barrier to accomplishing this goal, it would require utilizing motion records to estimate the surrounding environment with additional methods to estimate energy consumption, such as cardiograms. Recognizing the negative effect of the changing oceanic environment on seabirds (*Sydeman et al., 2015*), revealing the direct small-scale mechanisms of environmental factors

(such as wind, wave, tide, current, and sea surface temperature) effects on animal behavior, especially in take-off is urgently required. The concept of estimating the surrounding environment using motion records is a novel solution with great potential to unravel the small spatiotemporal uncertainties in seabird research.

## Materials and methods
### Field experiment
The recorders, Ninja-scan (Little Leonardo, Tokyo, Japan), record triaxial acceleration at a very high time resolution (100 Hz). Ninja-scan also records 3D GPS positions (5 Hz), Doppler velocity (5 Hz), temperature (6 Hz), pressure (6 Hz), geomagnetism (6 Hz), and angular velocity (100 Hz). There are two types of Ninja-scans with different battery masses (*Naruoka et al., 2021*). Small Ninja-scans weighed 28 g, which is 0.3–0.4% of wandering albatross body mass, and are expected to record for 7 hr. Large Ninja-scans weighed 91 g, which corresponds to 0.8–1.3% of wandering albatross body mass, and are expected to record for 65 h.

Ninja-scans were attached to breeding wandering albatrosses at Possession Island, Crozet archipelago (46°25 S, 51°44 E) in the South Indian Ocean in 2019 and 2020. In 2019, 12 small Ninja-scans were attached (in tandem) to 6 individuals. On each bird, one recorder had a delay timer so that the two recording periods did not overlap. Additionally, 15 birds had individual Ninja-scans attached, of which 8 were small Ninja-scans and 7 were large Ninja-scans. In 2020, 10 small Ninja-scans were attached in tandem to 5 individuals. Additionally, 19 birds had individual Ninja-scans attached, of which 7 were small Ninja-scans and 12 were large Ninja-scans. In summary, 21 and 24 wandering albatrosses were tagged in 2019 and 2020, respectively. All experiments were performed from late January to early March of each year, which corresponds to the incubation period of wandering albatrosses. Recorders were attached to the back of each bird with waterproof tape (Tesa, Hamburg, Germany) and glue (Loctite; Henkel, Dusseldorf, Germany). All recorders were retrieved within 35 days. One small Ninja-scan which had been attached in isolation in 2020 did not work correctly. The effects of the attached recorders on wandering albatrosses were previously assessed (*Phillips et al., 2003*; *Barbraud and Weimerskirch, 2012*) and revealed that small recorders (less than 3% of their body mass) do not negatively impact breeding or foraging behaviors. The experiment was conducted as part of Program 109 of the Institut Polaire Paul Emile Victor with permission from the Préfet des Terrs Australes et Antarctiques Françaises, France (permit numbers: 2018-117 and 2019-106).

### Take-off identification
First, data recorded on the colony island were eliminated based on the GPS position. Then, take-off was determined using the absolute value of the GPS horizontal velocity. When wandering albatrosses float on the sea surface (i.e., before take-off), a relatively low speed which is generally below 2.5 m/s, is recorded, while the flying speed exceeds 5 m/s (*Weimerskirch et al., 2002*). Take-off was defined as the moment when the horizontal speed exceeds 4 m/s and rises to a higher speed. The soaring (flying) speed occasionally meets this criteria. Therefore, the horizontal speed was smoothed using the moving average (20 points: 0.4 s). If the horizontal speed crossed the 4 m/s line several times within a short period, they were classed as take-offs for very short flights and were not used in our investigation. Therefore, we selected only take-offs that included over 30 s of floating followed by over 30 s of flying.

### Wind estimation
*Yonehara et al., 2016* proposed estimating the wind speed and direction of seabird flight paths using the sinusoidal curve relationship between flight speed and flight direction. When seabirds fly in the air, their flight speed against the ground (ground speed) is mainly affected by the wind speed, which is maximized in tail winds and minimized in headwinds. The maximum speed is the sum of the flight speed against air (air speed) and wind speed, whereas the minimum speed is the difference between the air speed and wind speed. The relationship between the flight speed (ground speed) and flight direction recorded by the GPS are fitted using a sinusoidal curve (*Shimatani et al., 2012*). We followed the methodology in *Yonehara et al., 2016* to collate the flight speed $V$ and flight direction $\theta$ data for 5 min after take-off and the curve was fitted using the following equation:

$$V = V_a + V_w \cos\left(\theta + \phi_w\right)$$

where $V_a$ is the air speed, $V_w$ is the wind speed, and $\phi_w$ is the wind direction. Ten seconds immediately after the take-off moment was not included in the estimation. Following *Yonehara et al., 2016*, the AIC of the sinusoidal fitting was compared to the linear fitting with a fixed slope of zero. When the AIC difference between the linear and sinusoidal fitting was below 2, the estimated results were considered unreliable and discarded. Wind speeds and directions were not calculated when take-offs were not followed by over 5 min of flight. The sinusoidal fitting was performed using Igor Pro version 8.04 (Wavemetrics, Portland, OR, USA).

## Wave estimation

The ocean wave properties experienced by seabirds before take-off were estimated by analyzing the floating motion at the sea surface (*Uesaka et al., 2022*). The wave height was estimated from the vertical GPS displacement records before take-off. The estimate requires sufficiently long records of vertical displacement. Therefore, the wave height was not calculated for take-offs that did not follow a surface floating time of over 15 min. The sampling period of 15 min ensured the reliability of the wave statistics (*Whitford et al., 2001*) and provided a large volume of estimated wave data. The estimate did not include the 10 s before the detected take-off moment. We followed the methodology of *Uesaka et al., 2022*. The vertical GPS displacement records were high-pass filtered using a cut-off frequency of 0.07 Hz to eliminate the GPS-derived error (*Olynik et al., 2002*). We separated the time series record of the vertical displacement into individual waves by applying the zero-up-crossing method. The mean wave height of the highest third of all individual waves was calculated to provide the significant wave height, which is the most widely used statistical wave parameter (*Whitford et al., 2001*).

## Sea surface running by seabirds

Many procellariiformes require a running phase before take-off from the sea surface (*Sato et al., 2009*). However, studies using accelerometers have not focused on the acceleration signal of this behavior. Surface running involves asymmetrical leg movements. Therefore, the lateral acceleration obtained from the recorder (attached to the back of the seabirds) provided signals derived from the running motion (*Figure 8*). We confirmed that running signals appear in the lateral acceleration records at the moment of take-off by streaked shearwaters (*Calonectris leucomelas*), which are phylogenetically similar to wandering albatrosses (see Supplementary Information Text and Figure S1).

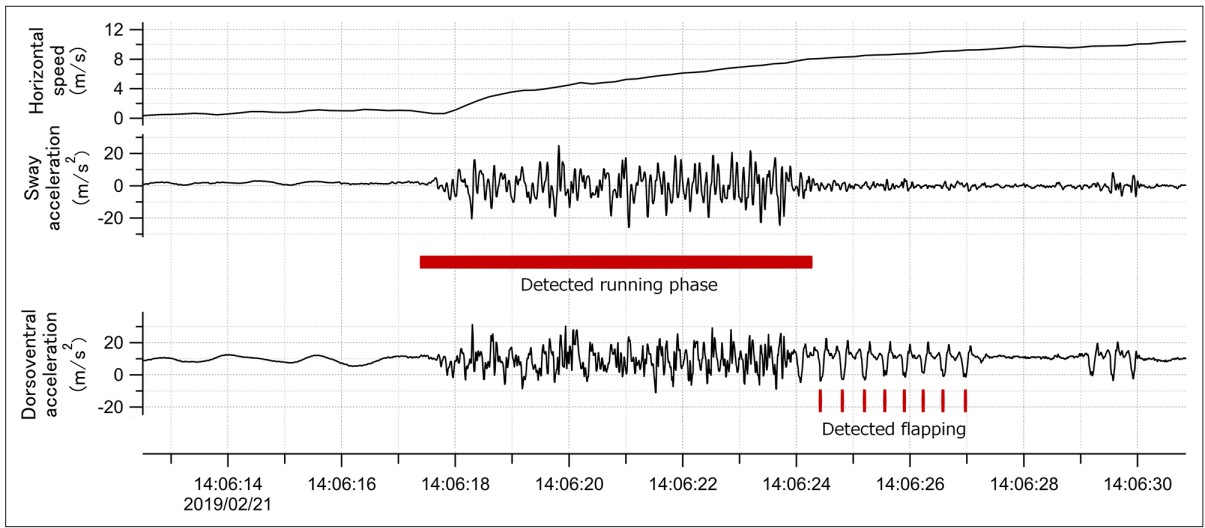

**Figure 8.** Time series data of horizontal speed (top), lateral acceleration (middle), and dorsoventral acceleration (bottom) signals of the wandering albatross at the moment of take-off. Horizontal speed starts increasing from the beginning of the take-off. Red square shows the detected running phase based on the variance of the lateral acceleration signal. Red bars show the detected flapping behavior after the running phase based on the dorsoventral acceleration signal. Dorsoventral signal during the running phase fluctuates, probably due to the shaking body derived from the running motion, and thus it is not easy to judge the existence of flapping behavior.

To explore the running duration of wandering albatross, we constructed an algorithm to detect the running phase from the lateral acceleration around take-off. The lateral acceleration signal is composed of a dominant component (0.25–0.4 s) and a high-frequency fluctuation component (<0.2 s period). Although the dominant component is the lateral movement derived from surface running, the flapping period of wandering albatross appears around this period (0.3–0.4 s). The flapping behavior is laterally symmetrical and does not appear in the lateral acceleration records. However, this is not always the case, when (occasionally) recorders are attached to the back of the seabird in a slightly tilted position. To avoid confusion between running and flapping behavior, a high-frequency fluctuation component in the lateral acceleration signal was used to detect the running phase. A band-pass filter was designed to extract the high-frequency fluctuation component from the acceleration records, and then the variance per unit time (0.6 s) was calculated at each point. Running phase was defined as when the acceleration variance exceeded the threshold value (2% of the peak value). This algorithm reasonably detects the running phase regardless of the running duration. If there is a signal gap in the middle of the running phase, the algorithm regards the gap as the end of the running phase, underestimating the running duration. However, these cases are rare, and we assume it does not affect our evaluation of the running characteristics of wandering albatross. The horizontal speed at the end of the running phase and take-off direction were calculated using the GPS velocity. The take-off direction was defined as the vectoral average direction during the running phase. We also calculated the cruising direction which was defined as the moving direction 5 min after take-off. All procedures were performed using Igor Pro version 8.04 (Wavemetrics, Portland, OR, USA).

## Flapping behavior after the running phase

Dorsoventral acceleration records include signals derived from seabird wing flapping behavior (*Figure 8*). The flapping signals during the running phase fluctuate, which is assumed to be caused by the leg-derived dorsoventral motion. This caused the flapping data to be unclear in identifying the flapping number and frequency. Therefore, we only focused on the wing flapping signals after the running phase. A band-pass filter extracted the clearest flapping signals (1.8–4.0 Hz). The number of continuous flapping signals after the running phase was counted. The flapping period of the wandering albatross is approximately 0.3–0.4 s, therefore we defined the end point when the flapping interval exceeded 0.5 s. The flapping frequency after the running phase was calculated using the spectral peak value of the continuous wavelet-transformed dorsoventral acceleration. All procedures were performed using Igor Pro version 8.04 (Wavemetrics, Portland, OR, USA).

## Comparison of the take-off parameters with environmental conditions

The wind directional bias of the take-off direction was tested using the *v*-test (modified Rayleigh test). The air speed $V_a$ at the end of the running phase was estimated using the following equation based on the parameters obtained from this study:

$$V_a = V_r + V_w \cos \left( \theta_t - \phi_w \right)$$

where $V_r$ is the running speed at the end of the running phase, $V_w$ is the wind speed, $\theta_t$ is the take-off direction, and $\phi_w$ is the wind direction. The effects of wind speed and wave height on each take-off parameter (running duration, running speed, flapping number, and flapping frequency) were evaluated using LMM with individuals treated as random effects. To identify significance levels, the models were compared to null models based on the AIC value.

To evaluate the combined effect of wind and waves, we categorized take-off conditions into four categories, 'WL conditions', 'WH conditions', 'SL conditions', and 'SH conditions'. Threshold values were decided based on the peak in the curve of the fitted probability density distribution (wind speed: 6.0 m/s, wave height 2.8 m). Weibull distribution and log normal distribution were used as the fitting function for wind speed and wave height, respectively (*Ferreira and Guedes Soares, 2000*; *Carta et al., 2009*). The values of each take-off parameter were compared between the four categories by Kruskal–Wallis test. Furthermore, the independent effects of wind and waves on take-off parameters were evaluated using LMM, including wind speed, wave height, and their interaction as explanatory parameters with individuals as random effects. VIF was also calculated before the LMM analysis to assess whether the multicollinearity effect could be dismissed. The *v*-test was performed using

Igor Pro version 8.04 (Wavemetrics, Portland, OR, USA). Statistical test and LMM calculations were performed using the Python 3.0 and PypeR package.

## Acknowledgements

We thank Yoshinari Yonehara, Julien Collet, Timothée Poupart, and all the members of the research station in the Crozet Islands for their support during the fieldwork. We appreciate the fruitful comments of Kagari Aoki, Chihiro Kinoshita, Laxmi Kumar Parajuli, and Aran Garrod. We are grateful to Taichi Sakamoto and Takashi Mukai (ATTACCATO) for providing customized rechargeable Li-ion batteries for the Ninja-scans. We also thank Michihiko Suzuki and Koichiro Ikeda (Little Leonardo) for molding the Ninja-scans. The study was financially supported by Grants-in-Aid for Scientific Research from JSPS (22K21355 to K Q Sakamoto). The fieldwork was funded by IPEV program n°109, and financially supported by a research project entitled 'Cyber Ocean: next generation navigation system on the sea' funded by the CREST program (JPMJCR1685) of Japan Science and Technology Agency; Grants-in-Aid for Scientific Research from JSPS (17H00776 to K Sato); the Tohoku Ecosystem-Associated Marine Science; Sasakawa Scientific Research Grant from The Japan Science Society (2020-4034 to L Uesaka); European Community's Seventh Framework Program FP7/2007–2013 funded by a European Research Council (ERC-2012-ADG_20120314 to H Weimerskirch); European Community's H2020 Program funded by the European Research Council (ERC-2017-PoC_780058 to H Weimerskirch).

## Additional information

### Funding

| Funder | Grant reference number | Author |
|---|---|---|
| Japan Science and Technology Agency | CREST program (JPMJCR1685) | Katsufumi Sato |
| Japan Society for the Promotion of Science | Grants-in-Aid for Scientific Research (17H00776) | Katsufumi Sato |
| Japan Society for the Promotion of Science | Grants-in-Aid for Scientific Research (22K21355) | Kentaro Q Sakamoto |
| Ministry of Education, Culture, Sports, Science and Technology | Tohoku Ecosystem-Associated Marine Science | Katsufumi Sato |
| Japan Science Society | Sasakawa Scientific Research Grant (2020-4034) | Leo Uesaka |
| European Research Council | European Community's Seventh Framework Program FP7/2007-2013 (ERC-2012-ADG_20120314) | Henri Weimerskirch |
| European Research Council | European Community's H2020 Program (ERC-2017-PoC_780058) | Henri Weimerskirch |
| Polar Institute Paul-Emile Victor | IPEV program n°109 | Henri Weimerskirch |

The funders had no role in study design, data collection, and interpretation, or the decision to submit the work for publication.

### Author contributions

Leo Uesaka, Conceptualization, Data curation, Software, Formal analysis, Visualization, Methodology, Writing - original draft; Yusuke Goto, Resources, Writing – review and editing; Masaru Naruoka, Resources, Software, Writing – review and editing; Henri Weimerskirch, Katsufumi Sato, Funding acquisition, Project administration, Writing – review and editing; Kentaro Q Sakamoto, Resources, Supervision, Project administration, Writing – review and editing

### Author ORCIDs

Leo Uesaka (iD) http://orcid.org/0000-0002-6703-696X
Masaru Naruoka (iD) http://orcid.org/0000-0002-0043-279X

### Ethics

Animal care was performed humanely following rules issued by the Réserve Nationale des Terres Australes. The field procedures and manipulations on Crozet, after approval from Comité National de la Protection de la Nature (CNPN), were given permission by the 'Préfet of Terres Australes et Antarctiques Françaises', permit numbers: 2018-117 and 2019-106. The effects of the attached recorders on wandering albatrosses were previously assessed (Phillips et al., 2003; Barbraud and Weimerskirch, 2012) and every effort was made to minimize the negative impacts on albatrosses.

Reviewer #1 (Public Review): https://doi.org/10.7554/eLife.87016.3.sa1
Reviewer #2 (Public Review): https://doi.org/10.7554/eLife.87016.3.sa2
Reviewer #3 (Public Review): https://doi.org/10.7554/eLife.87016.3.sa3
Author Response https://doi.org/10.7554/eLife.87016.3.sa4

## Additional files

### Supplementary files

• Supplementary file 1. Information on individuals and attached recorders.

• MDAR checklist

### Data availability

Raw positional and motion data of wandering albatrosses have been deposited in Dryad and Biologging intelligent Platform (BiP). Anyone can see figures of all positional data in BiP. Creating an account is required to download data and metafiles. Ocean wind and wave conditions can be also directly calculated from the raw positional and motion data using an embedded OLAP system in BiP. The original code to reproduce behavioral data, environmental data, and figures has been deposited to GitHub (copy archived at *Uesaka, 2023*).

The following dataset was generated:

| Author(s) | Year | Dataset title | Dataset URL | Database and Identifier |
|---|---|---|---|---|
| Uesaka L, Goto Y, Naruoka M, Weimerskirch H, Sato K, Sakamoto KQ | 2023 | Behavioral datasets of wandering albatrosses collected at Possession Island, Crozet, France, in 2019 and 2020 | https://doi.org/10.5061/dryad.tx95x6b2j | Dryad Digital Repository, 10.5061/dryad.tx95x6b2j |

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

## Appendix 1

### Running signal confirmation using streaked shearwater

Among ecological studies using animal-borne recorders, accelerometers are often used to detect specific animal behaviors. The running behavior of seabirds (which is required when they take off from the sea surface) should be a specific feature of the acceleration signal. Therefore, we confirm the existence of a running signal in the acceleration record using streaked shearwaters (*Calonectris leucomelas*).

Streaked shearwaters belong to the same phylogenetic group as wandering albatross, Procellariiformes, and have a similar body silhouette, although the body mass is 20 times lighter than wandering albatross. We attached a combined video and acceleration recorder (DVL400–3DGT, Little Leonardo, Japan) to the chest of streaked shearwaters breeding at Funakoshi Ohshima Island (39°24′N,141°59′E), Japan, during the chick-rearing period from August to September 2021. Recorders were attached to the chest of each bird using waterproof tape (Tesa, Hamburg, Germany) and glue (Loctite; Henkel, Dusseldorf, Germany) with the camera facing backward to film leg movement. The running duration was filmed by the video, and the sway acceleration signals were compared. The time resolution of the acceleration records was 100 Hz. All procedures were approved by the Animal Experimental Committee of Atmosphere and Ocean Research Institute, University of Tokyo (A21-10), and were conducted with permission from the Ministry of the Environment and Agency for Cultural Affairs, Japan.

At the take-off moment, streaked shearwaters move their legs, as demonstrated by the video record (*Appendix 1—figure 1*). The simultaneous record of sway acceleration showed specific signals during the running phase (*Appendix 1—figure 1*). The dominant period of the signal was approximately 0.16–0.20 s (*Appendix 1—figure 1*). The running leg cycle was analyzed using a captured bird, the cycle was approximately 0.19 s which corresponds to the dominant period of the sway acceleration signal. We conclude the signal of the sway acceleration record is derived from the running motion of seabirds.

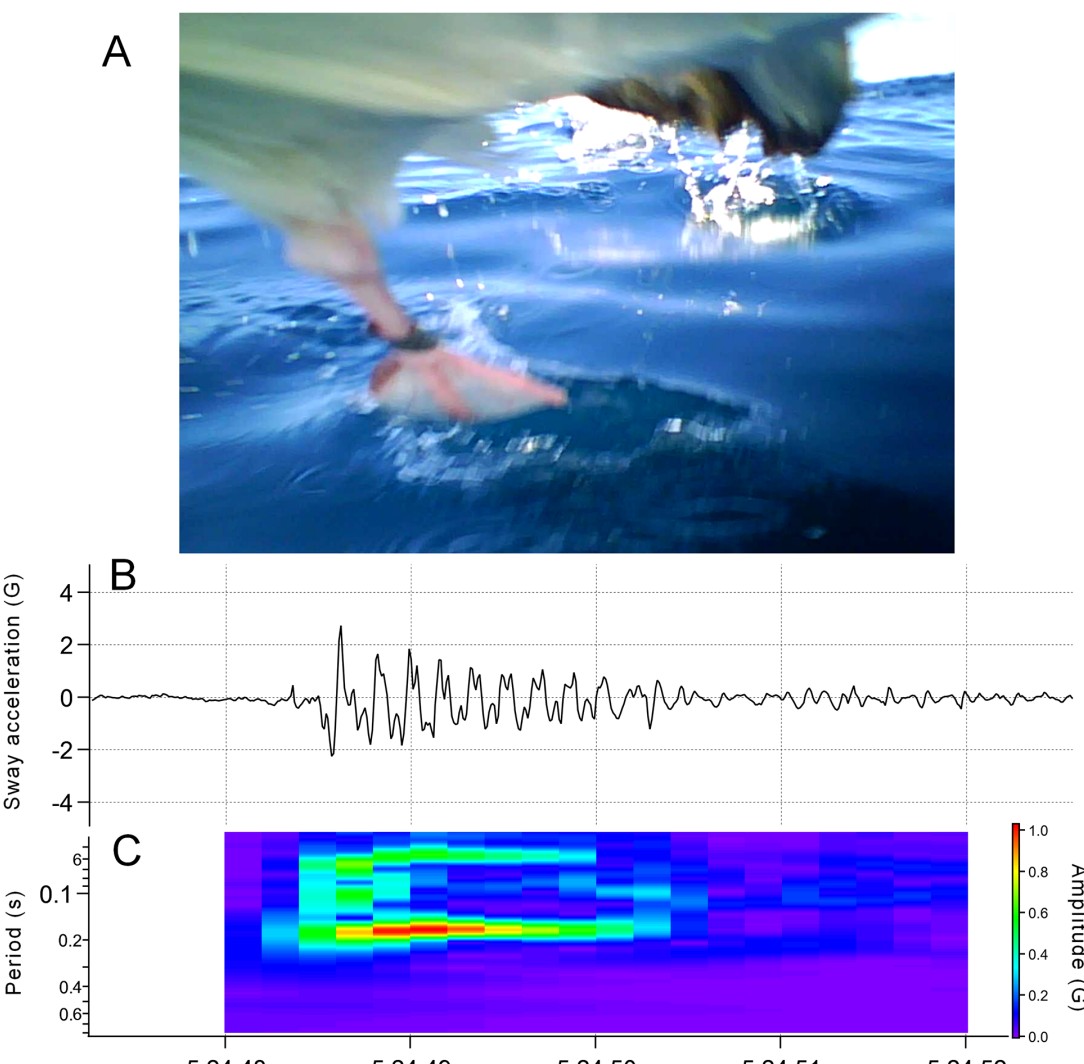

**Appendix 1—figure 1.** Leg motion and acceleration signal during seabird take-off. (**A**) The video recorder mounted on the chest of the bird captured their legs moving in turns to run on the sea surface. (**B**) Sway acceleration record of taking-off streaked shearwater. Specific signal emerges only when their legs are moving. (**C**) Dominant period of the signal calculated by continuous wavelet transform using Igor Pro 8 (Wavemetrics, Portland, OR, USA).

