## [Editor Report · eLife assessment]

This **fundamental** study advances our understanding of seabird responses to environmental conditions, with implications for movement ecology, flight biomechanics, animal foraging, and bio-energetics. Animal-borne data-loggers are used to generate a **compelling** high quality dataset on animal movement and environmental conditions. The study will interest ornithologists, comparative bio-mechanists, ocean ecologists and those interested in technological advances in animal sensors.

---

## [Referee Report · Reviewer #1 (Public Review)]

The authors were seeking to improve understanding of how wind and wave action affect the use of energetically demanding wing flapping and running by albatross engaged in takeoff flight. To accomplish this in the complex and challenging environment in which albatross live, the authors sought to use accelerometry and geographic positioning to infer patterns of locomotion, flight orientation relative to the prevailing wind, and wave height during takeoff.

The major strength of the methods and results is that the use of accelerometry and novel interpretations of data from a geographic positioning system provides new insight into the use of waves by albatross and how the effects of wave magnitude interact with the wind to modulate energy demands during takeoff. Weaknesses of the approach are due to the challenging environmental conditions in which albatross live. The interpretation of accelerometry data was not validated using a subset of the sample synchronized with video (prior validation was cited for shearwaters). The interpretation of wind direction relative to flight path is based on the behavior of the bird without concurrent measures of local wind velocity.

The authors achieved their aims, and their results support their conclusions.

Although it is generally understood that albatross and many other birds choose to takeoff into the wind to reduce energetic costs, the authors provide novel quantitative data on this behavior. Their results on the effects of wave height and the interactions between wave height and wind provide novel insight into how albatross harvest energy from their complicated and dynamic environment to reduce the energy they must output to get into the air. In particular, the new insight into the effects of wave height should revise understanding among ornithologists, ocean ecologists and those who study the mechanics of animal locomotion. The use of accelerometry and geographic positioning systems to measure flight behavior and ocean ecology should inspire other researchers to adopt similar methods.

Albatross live in a complex and poorly understood environment that is likely to be threatened by climate change. This research provides worthwhile new insight into how wind and wave action affect takeoff in albatross, and can therefore improve insight into how changes in these variables with climate change may affect the distribution of albatross populations.

---

## [Referee Report · Reviewer #2 (Public Review)]

The authors used cutting-edge bio-telemetry technology to decipher the roles of wind speed and wave height on the take-off of albatrosses from the water surface. They revealed that each of these factors contributes to take-off in a unique way with interesting interactions of the two factors. The authors achieved their objectives and their results support their conclusions. This work will set new standards in integrating information about bird movement and environmental conditions experienced by the bird in a comprehensive, integrative and hypothesis-driven framework. The approach of the authors is highly advanced, providing heuristic insights for many additional systems where organisms are influenced by, and respond to small-scale environmental conditions.

---

## [Referee Report · Reviewer #3 (Public Review)]

The present study used novel data logging devices to record the foraging behavior of wandering albatrosses. Specifically, the authors showed how localized winds and wave heights influence their ability to take off from the sea surface, which is the most expensive behavior they engage in while foraging. There is no better platform for this initial work because these birds are so large, the equipment they can carry without creating significant impact is tremendous.

The results were impressive, presented well, and the paper was generally written in an accessible way to readers with less knowledge. The authors provide a convincing set of results that support the aims and conclusions. The theory and application could be used to inform our understanding of flight behavior in other seabirds.

Although the idea of taking off from the sea surface may sound trivial, it is essential to understand that albatrosses and other soaring seabirds have wings that are adapted for soaring (i.e. long and narrow). The trade off, however, is that powered flight through wing flapping is energetically expensive because the wings have a shallow amplitude and generate less power compared to a shorter, wider wing. Thus, wind is everything and this study shows how wind facilitates the ability of the birds to gain flight using wind and waves. Awesome!

---

## [Author Response]

The following is the authors’ response to the original reviews.

We appreciate the thoughtful feedback provided by the editor and the three reviewers and have addressed their comments, which we believe have results in significant improvements to the manuscript. A point-by-point response to the comments is included below.

**Reviewer #1**
Line 229: The wording of "highly valuable" seems slightly vague. Consider rephrasing to something more specific, such as: "...using individual animal recorders provide valuable new insight into locomotor behavior when ..."

Thank you for your advice. The sentence was revised as you suggested.

Lines 518-527: Consider adding quantitative details for the four conditions. It is apparent in Figure 4 (dashed lines associated with peaks of the distributions), but in the text it would be helpful to add the speeds and heights chosen to sort the data.

Quantitative descriptions were added in the Materials and Methods section. We also moved detailed information about the curve fitting function from the Result section to Materials and Methods section.

“Threshold values were decided based on the peak in the curve of the fitted probability density distribution (wind speed: 6.0 m/s, wave height 2.8 m). Weibull distribution and log normal distribution were used as the fitting function for wind speed and wave height, respectively (Ferreira and Guedes Soares, 2000; Carta et al., 2009).”

**Reviewer #2**
Line 51 - Climatic models - climatic model cannot, by definition, provide prediction of specific weather conditions as they focus on large and long-term values and trends. I suggest the authors to review their use of climatic conditions throughout the manuscript, and use instead weather conditions, where appropriate.

Thank you for informing us the usage of terminology. Most of the phrases “climatic models” in the manuscript were replaced by “mathematical weather models”, for example, line 51, 226, 230, 312. We also checked that the phrase “climatic condition” never appears in the manuscript.

Lines 59-61 require editing. It is true that take-off is associated with high rate of energy expenditure, but it is phrased in an unclear way. I suggest writing instead "Therefore, the high energy expenditure associated with take-off is strongly influencing the total energy expenditure of wandering albatross during the foraging trip, unlike the duration or distance of the flight (Shaffer et al., 2001).

Thank you for your advice. As you suggested the phrasing was not proper to describe the previous study. The sentence was revised following your suggestion.

“Therefore, the high energy expenditure associated with take-off strongly influences the total energy expenditure of wandering albatross during the foraging trip, unlike flight duration or distance (Shaffer et al., 2001a)”

Line 213 - I suggest "Among the LMMS, models..."

The sentence was revised as you suggested.

Line 286 - I suggest using the word "difference" or "delta" AIC instead of variation which is confusing.

The sentence was revised as follows.

“For instance, the AIC difference in running speed between the best model and the second-lowest AIC model was only 0.27.”

Line 385 - Please provide actual percentage even if it is < 1%.

We added actual mass percentage of both small and large types of the recorders in line 385 and 386.

“Small Ninja-scans weighed 28 g, which is 0.3 ~ 0.4% of wandering albatross body mass, and are expected to record for 7 h. Large Ninja-scans weighed 91 g, which corresponds to 0.8 ~ 1.3% of wandering albatross body mass, and are expected to record for 65 h.”

**Reviewer #3**

Thank you for the marked-up manuscript and a lot of comments on it. Most of your grammatical advises and rephrases are reflected in the new version and we double-checked the whole manuscript using English proofreading service. Please refer to the below for the answer to each major comments.

Line 304 – not sure volume is best word choice.

We changed the word “volume” to “amplitude”.

Line 309 – Are you sure that Pennycuick 1982 didn’t document this?

His article mainly focused on the morphology and steady flight mechanism of albatrosses and petrels. There were no descriptions on take-offs of seabirds.

Line 320 – add after Weimerskirch citation, and similar to predicted best glide speeds (Shafer et al. 2001, Funct, Ecol 15)

Thank you for the beneficial information. We added the phrase and the citation in the sentence.

“The mean air speed of wandering albatrosses at the end of the running phase was close to the average flight speed (approximately 15 m/s) (Weimerskirch et al., 2002), and similar to predicted best glide speeds, (Shaffer et al., 2001b) indicating that wandering albatrosses gain sufficient lift at the end of the running phase and efficiently utilize ocean wind.”

Line 684 – citation information incomplete.

Thank you for finding the incomplete citation. Authors of the reference paper were corrected.“Weimerskirch H, Bonadonna F, Bailleul F, Mabille G, Dell’Omo G, Lipp H-P. 2002. GPS tracking of foraging albatrosses. Science 295:1259–1259. doi:10.1126/science.1068034”

Fig.4. – In Part B, reorient the y-axis labels to match the other figures. Change the orientation of y-axis labels like shown in Figure 3.

We rearranged the labels and ticks in Fig.4B to improve the readability and match the graphs with Fig3.